# Precision Oncology with Drugs Targeting the Replication Stress, ATR, and Schlafen 11

**DOI:** 10.3390/cancers13184601

**Published:** 2021-09-14

**Authors:** Ukhyun Jo, Yasuhisa Murai, Naoko Takebe, Anish Thomas, Yves Pommier

**Affiliations:** 1Developmental Therapeutics Branch and Laboratory of Molecular Pharmacology, Center for Cancer Research, NCI, NIH, Bethesda, MD 20892-4264, USA; murai.y@hirosaki-u.ac.jp (Y.M.); anish.thomas@nih.gov (A.T.); 2Department of Gastroenterology and Hematology, Hirosaki University Graduate School of Medicine, Hirosaki 036-8562, Japan; 3Developmental Therapeutics Branch and Division of Cancer Treatment and Diagnosis, NCI, NIH, Bethesda, MD 20892-4264, USA; takeben@mail.nih.gov

**Keywords:** replication stress, ATR, schlafen 11, berzosertib, Camptothecin, cisplatin, Ceralasertib, Adavoserib, PARP, olaparib, niraparib, talazoparib, Rucaparib, temozolomide

## Abstract

**Simple Summary:**

Chemotherapeutic DNA-damaging agents targeting replication are widely used but predictive rationales for drug combinations and patient selection still need clinical definition. Here, we review cancer-associated replication stress (RepStress) and its genomic signature, and propose how to utilize RepStress-targeted therapies in the context of ATR inhibitors and Schlafen 11 (SLFN11).

**Abstract:**

Precision medicine aims to implement strategies based on the molecular features of tumors and optimized drug delivery to improve cancer diagnosis and treatment. DNA replication is a logical approach because it can be targeted by a broad range of anticancer drugs that are both clinically approved and in development. These drugs increase deleterious replication stress (RepStress); however, how to selectively target and identify the tumors with specific molecular characteristics are unmet clinical needs. Here, we provide background information on the molecular processes of DNA replication and its checkpoints, and discuss how to target replication, checkpoint, and repair pathways with ATR inhibitors and exploit Schlafen 11 (SLFN11) as a predictive biomarker.

## 1. Introduction

Targeting the genome and DNA replication were the first approaches to the chemical treatment of cancers. Most of the drugs were discovered empirically and later found to have highly specific targets and molecular mechanisms of action [1]. For example, DNA topoisomerase (TOP) inhibitors were discovered as potent anticancer agents prior to the elucidation of their molecular mechanism of action by selectively trapping of the TOP cleavage complexes (TOPcc) by interfacial inhibition [1,2,3]. DNA replication is widely targeted in the anticancer armamentarium by various replication stress (RepStress) inducers including TOP inhibitors, alkylating agents, platinum compounds, and poly (ADPribose) polymerase (PARP) inhibitors [4,5,6]. While RepStress inducers cause DNA damage directly, additional targeting of the replicative stress response signaling pathways is emerging as a potential combination strategy by which detrimental RepStress and irreparable DNA breakage accumulate [7,8]. For the clinical success of these RepStress-targeted therapies, identification of reliable biomarkers is essential. For instance, the selection of patients according to the expression of Schlafen 11 (SLFN11) has recently been reported to be correlated with improved response in several human cancer types [9,10,11].

Here, we review precision medicine in the context of RepStress-targeted agents, which primarily act by interfering with DNA replication. We summarize key molecular features of DNA replication and discuss how to exploit the RepStress, ATR (ataxia telangiectasia mutated and Rad 3-related) protein kinase inhibitors and SLFN11 for cancer therapy. Figure 1 outlines the molecular mechanisms of action of SLFN11 and how they differ from ATR; they are detailed in Section 5.1.

## 2. Precision Medicine

Precision medicine is rapidly expanding for cancer diagnosis, treatment, and prognosis [12] providing a roadmap to classify cancer patients into sub-groups that differ in their susceptibility to a particular tumor incidence and evolutive outcome, or in their response to anticancer therapies (Figure 2). Knowledge of this framework should enable cancer patients to receive the most effective and safest treatments, while avoiding the use of ineffective therapies with toxic side effects.

The paradigm for precision medicine is exemplified by the MATCH (molecular analysis for therapy choice) clinical trials with emphasis on kinase inhibitors and phase 0 trials [13,14,15,16,17,18,19]. The identification of predictive and pharmacodynamic biomarkers has also been a recent focus for the development of immune checkpoint inhibitors [20] as they yield durable clinical benefit in relatively small numbers of patients, while producing potentially severe toxicity and high financial burden for a large number of non-responding patients [21].

Extending precision medicine to the large number of widely used agents targeting the genome and DNA replication remains a mostly unmet need. Completion of the Human Genome Project has paved the way for groundbreaking advances in sequencing technologies that have profoundly impacted precision medicine by providing access to the underlying genetic codes of cancer development and progression [22]. The combination of next-generation sequencing with high-throughput analyses is also providing massive information to elucidate the cancer genomes. In parallel, new drugs have been developed for specific molecular targets. Trastuzumab and imatinib have opened the gate for targeted cancer therapy based on tumor genetic characteristics [23]. They are routinely used to treat patients with HER2-amplified metastatic breast cancer and *BCR*-*ABL* fusion-positive chronic myelogenous leukemia, respectively. However, omics analyses remain technically complex and require a bioinformatic infrastructure limited to only a small number of cancer centers. In addition, genetic mutations provide a limited spectrum of therapeutic options, and it is increasingly clear that transcriptome analyses based on whole genome RNA sequencing, gene copy number determinations, epigenetic and proteomic analyses need to be developed and included (part 2 in Figure 2). For instance, lack of methylguanine methyltransferase (MGMT) expression predicts potential response to temozolomide [24,25] and high Schlafen 11 (SLFN11) expression potential response to DNA replication-targeted therapies (Table 1) [11].

Yet, single gene features (mutations and expression) are generally insufficient for predicting drug response. Multivariate analyses and artificial intelligence (AI) approaches are likely to increase the predictive value of omics analyses. For instance, cancer cells expressing high SLFN11 but with high expression of the drug efflux genes ABCB1 (encoding P-glycoprotein) are unlikely to respond to doxorubicin [26]. Multivariate analyses can be combined to provide molecular signatures such as mismatch repair (MMR), homologous recombination (HR), neuroendocrine (NE), epithelial mesenchymal (EMT) signatures (part 3 in Figure 2). Moreover, molecular analyses have led to the concept of synthetic lethality with the use of poly (ADP ribose) polymerase (PARP) inhibitors for BRCA- and HR-deficient tumors [27,28] (part 4 in Figure 2).

Ultimately, precision medicine can be achieved by concentrating the therapeutic delivery to tumors while sparing normal tissues, as is the case for surgery and radiotherapy. This goal is now achievable with tumor-targeted antibody drug conjugates (ADC) that use DNA-targeted payloads such as the DNA alkylating agent PBD (pyrrolobenzodiazepine dimer) [29] or the TOP1 inhibitors, SN-38 and deruxtecan [30] (part 5 in Figure 2). Finally, targeting the tumor microenvironment (TME) with immune checkpoint inhibitors that block the programmed cell death-1 (PD-1) and cytotoxic T-lymphocyte associated protein 4 (CTLA4) pathways is becoming routine and can complement the above approaches for precision medicine (part 6 in Figure 2).

## 3. DNA Replication, Replication Stress (RepStress), and Maintenance of Genome Stability

Genome instability is a hallmark of cancer [31,32]. It is not only a driving force for tumorigenesis but also a key target of anti-cancer agents. These include not only the recently developed agents targeting Poly (ADP-ribose) polymerase (PARP) and ataxia telangiectasia and Rad3-related protein (ATR), but also the classical DNA-targeted chemotherapeutic agents such as antimetabolites, alkylating agents, platinum derivatives, and TOP inhibitors (Table 1). Among the various cellular processes, DNA replication is essential for preserving genome integrity. Its deregulation caused by exogenous or endogenous stresses results in replication fork collapse and DNA damage, frequently driving cancer development [4].

Many clinical anticancer drugs inhibit DNA replication by generating replicative stress through a range of mechanisms, ultimately inducing genomic damage and killing the cancer cells [33] (Table 1). However, molecular surveillance and repair machineries in cancer cells hinder the therapeutic effects of RepStress-targeted anticancer agents through DNA lesion detection, repair, and cell cycle control [34]. The replicative DNA damage response (DDR) pathways are among the well-defined cellular processes in initiation, sensing, and correction of DNA lesions. Thus, understanding the DNA replication-targeted agents and DDR is required for providing biomarkers (Figure 2) that can be utilized for selecting patients who should benefit from targeted therapies.

### 3.1. Overview of DNA Replication

The coordination of DNA replication is highly regulated in a timely manner. DNA replication is also an orchestrated and dynamic process consisting of multi-subunit protein complexes for the accurate timing of replication initiation, elongation, and termination, chronologically encompassing the G1, S, and G2 phases of the cell cycle (Figure 3).

In G1 where chromosomes are relaxed, replication is initiated by the assembly of replication initiation factors on replication origins scattered along chromosomes [4,35]. Remarkably, only a small subset of replication origins is licensed by the loading of the six-subunit origin recognition complex (ORC), the cell division cycle 6 (CDC6), the chromatin licensing, and DNA replication factor 1 (CDT1). This follows the recruitment of the replicative DNA helicase minichromosome maintenance complex 2-7 (MCM2-7) as the pre-replicative complex (pre-RC) that remains in an inactive state prior to S-phase. Subsequentially, CDC6 and CDT1 are released from the initiated replicons to prevent the re-licensing of same replication origins [36,37]. Indeed, the licensing of replication origins at pre-RC needs to occur restrictively in the G1-phase [38]. Otherwise, it can cause replication restart from already fired replication origins during S-phase, which leads to replication catastrophe and DNA breakage [39]. Cyclin-dependent kinases (CDKs) are key regulators controlling new origin firing from late G1 phase to mitosis to ensure the duplication of the entire genome only once per cell division.

In S-phase where most of the genome is duplicated, the MCM helicases trigger origin firings by their phosphorylation mediated by the CDKs and the Dbf4-dependent kinase (DDK), subsequentially recruiting CDC45 and GINS within CMG active helicase complexes [40]. At the same time, some of the MCM2-7 complex on replication origins remain dormant. Yet, these dormant origins can be activated under RepStress for compensating main origins have given rise to aborted replicons [36,41]. The activated CMG helicase complex unwinds and separates the DNA double-helix into single-stranded DNA, forming the classical Y-shaped replication forks (Figure 3). Once the DNA double-strands are separated, replication forks spatially recruit a large set of proteins to form replisomes with DNA polymerases, helicases, PCNA (proliferating cell nuclear antigen), primase, RNase H, ligases, and TOPs [42]. DNA synthesis is initiated by DNA polymerase α from RNA primers that are replaced with DNA at the end of DNA replication. DNA polymerases ε and δ extend the leading and lagging strands, respectively (Figure 3). PCNA plays an essential role in replication as processivity factor for the polymerases and for switching the DNA polymerases upon RepStress and for translesion synthesis (TLS) [43].

DNA replication is terminated in G2-M by the convergence of replication forks [44]. At the end of the S- and in G2-phases, all remaining damage in the newly replicated DNA caused by RepStress must be resolved to ensure that the whole genome information is completely duplicated before the chromosomes are segregated during mitosis.

### 3.2. Replication Stress (RepStress)

Because of their molecular complexity and dynamism, replication machineries are prone to encounter various stress conditions derived from spatial and temporal obstacles including dNTP pool depletion, collisions between replication forks and/or transcription, DNA single- and double-strand breaks (SSBs and DSBs), unscheduled firing of dormant origins, topological stress resulting from DNA torsional tensions and condensed heterochromatin regions [4]. Additionally, replication can be hindered by DNA secondary structures that interfere with the loading of replication proteins and/or opening of the double helix, including mismatched DNA bases, misincorporated ribonucleotides, abnormal DNA structures such as guanosine quartets, palindromic cruciforms and knots, as well as DNA-DNA and DNA-protein crosslinks, which can all impair replication fork progression. Furthermore, shortage of deoxyribonucleotides (dNTPs), core proteins comprising the replication machinery, and the uncoupling of regulatory factors from replisomes can also interfere with replication. They strongly disturb DNA synthesis and replication progression, resulting in RepStress with fork stalling and collapse, single-ended DNA double-strand breaks (seDSBs), and unscheduled termination of replication [33].

Once replication forks are stalled, replication protein A (RPA) complexes accumulate on long stretches of single-stranded DNA (ssDNA), eventually causing global RPA exhaustion that leads to cellular catastrophe and DNA breakage [33,45]. These RPA filaments are key for the activation of the ATR and the SLFN11 response pathways (Figure 3).

### 3.3. Endogenous RepStress

“Sustaining proliferative signaling” is one of the six primary hallmarks of cancers highlighted by Weinberg and Hanahan [46]. To induce sustained proliferation, cancer cells rely on three main genetic drivers: (i) activation of oncogenes such as *MYC*, (ii) inactivation of tumor suppressors such as *TP53* and *RB1*, and (iii) a mutator phenotype related to reduced replication fidelity such as DNA polymerase alterations [47]. Additionally, oncogene-induced unscheduled firing of dormant origins and failure in sustaining DNA damage response and DNA repair lead to genomic instability including DNA hypermutations, gene deletions and amplifications, loss of heterozygosity, chromosomal rearrangements, and chromosome gain or loss (aneuploidy), which all contribute further to cancer progression and chemoresistance.

### 3.4. RepStress Induced by Clinically Approved Chemotherapeutic Agents

Many conventional chemotherapeutic agents are RepStress inducers (Table 1). They act by multiple mechanisms including depletion of dNTP pools or generation of cytotoxic DNA obstacles such as mismatches, abasic sites, breaks, DNA-protein crosslinks (DPCs) and DNA-DNA crosslinks that cause replication fork stalling. Direct inhibition of the replicative polymerases by chain terminating drugs such as cytarabine is another mechanism for stalling replication forks.

The various DNA polymerases carry out DNA synthesis with high proofreading ability and participate in replication-coupled repair processes [48]. Thus, blocking polymerase activity can be a pivotal therapeutic strategy for cancer treatment. In general, the activity of the DNA polymerases (Pol α, Pol δ, and pol ε) that carry out DNA synthesis is blocked by the incorporation of nucleoside analogs [49] (Table 1). Deoxycytidine nucleoside analogs target DNA polymerases by competing with dCTP, thereby resulting in failure of elongation of the nascent DNA strands [50]. Cytarabine (cytosine arabinoside [Ara-C]) specifically shows enhanced cytotoxicity in the BER enzyme 8-oxoguanine DNA glycosylase-1 (OGG1)-deficient acute myeloid leukemia cells [51]. A functional genomic screen in mesothelioma uncovered that loss of function of BRCA1-associated protein 1 (BAP1) is associated with vulnerability to ribonucleotide reductase (RNR) inhibitors such as gemcitabine [52].

Alkylating agents and platinum-based compounds used to treat a wide range of solid tumors directly attacking the DNA bases forming covalent DNA adducts that interfere with the progression of DNA polymerases. Expression of O-6-methylguanine-DNA methyltransferase (MGMT) is a commonly used biomarker for the alkylating agent temozolomide in glioma and glioblastoma patients [25], and temozolomide has recently shown to be effective in combination with the base excision repair inhibitor TRC102 (methoxamine) in patients with relapsed solid tumors and lymphomas [53] (Table 1). The sensitivity of platinum compounds including cisplatin and carboplatin is increased in triple-negative breast and serous ovarian cancers with overexpression of the Bloom (BLM) helicase [54]. Besides, BRCA-deficient cells are highly vulnerable to treatment of platinum-based compounds [55,56] and the TOP inhibitors [57,58].

During DNA replication, TOPs transiently cut DNA strands to resolve topological problems ahead of and behind replicating forks [59]. TOP inhibitors trap TOP cleavage complexes (TOPccs), leading to RepStress and DNA damage [30,59]. Recent studies found that homologous recombination defects (HRD and BRCAness) and SLFN11 expression increase susceptibility to TOP1 inhibitors [10,11,30] (Table 1). Additionally, the TOP1 inhibitors are currently being exploited as toxic payloads in tumor-targeted drug delivery by using liposomes, PEGylation, and antibody drug conjugates (ADC) to limit their toxic side-effects by selectively targeting tumor cells (part 4 in Figure 2) [30].

In a similar manner as TOP trapping [2,28], the most potent PARP inhibitors such as talazoparib and niraparib trap PARP1 on pre-existing DNA lesions along with their inhibitory activity of PARP, making them potent RepStress inducers [28,60]. The therapeutic benefits of PARP inhibitors are significantly associated with HRD cancers and expression status of SLFN11 (Table 1) [61,62].

Enhancing replicative damage by blunting the RepStress response pathways with ATR inhibitors is discussed below [8,63,64,65,66,67]. Synthetic lethality for the PARP inhibitors and platinum derivatives in HRD cancers is being applied for cancer treatment [68,69] and recent studies in animal models suggest such synthetic lethality for TOP1 inhibitors [10,58]. The PARP inhibitors, the TOP1 inhibitors, and platinum derivatives are also highly synergistic in combination with ATR and CHK1 inhibitors.

### 3.5. RepStress Induced by Non-FDA-Approved Drugs Targeting Replication

Aphidicolin, a natural tetracyclic alkaloid widely used in preclinical models, and which acts by blocking the incorporation of dCTP through its binding at the interface of the Pol α active site and rotating the template guanine [70], is not used clinically because of its poor pharmacokinetics [70]. Recently, a small molecule inhibitor of Pol δ (POLD) (Zelpolib) has been found to bind to the active site of the polymerase, inhibiting DNA replication and demonstrating synergy with the PARP inhibitors [71]. A small molecule inhibitor (PNR-7-02) of Pol η (POLH) that functions in DNA (TLS) also shows synergistic effects with cisplatin in HAP1 chronic myelogenous haploid leukemia cells [72] and inhibitors of the microhomology end-joining (MHEJ) polymerase, pol θ (POL Q) are being developed to overcome resistance to the PARP inhibitors [73,74]. Additionally, the retinoic-acid-derived inhibitor of pol α (POLA1), CD437 (AHPN), and ST1926 were recently reported to exhibit anticancer activity [75].

Given that the replicative MCM2-7 helicase complex plays a critical role by unwinding the DNA helix in the replisome during replication initiation and elongation (Figure 3), efforts to develop anticancer agents targeting the MCM helicase protein complex are ongoing [76,77,78]. Post-translational modifications of PCNA and inhibitory interactions with key regulators through their PCNA interacting protein box (PIP-Box) motif are being pursued to design anticancer drugs [79,80]. A small molecule inhibitor (T2AA: T2 amino alcohol) targeting monoubiquinated PCNA and inhibiting DNA replication and TLS significantly increases cisplatin-induced DSBs [81]. Another PCNA inhibitor PCNA-I1S was found to selectively bind PCNA, suppress its chromatin association, and to suppress cancer cell growth [82,83]. Furthermore, PCNA is functionally engaged with the stability of replication modulator proteins for DNA synthesis and replicative DNA damage response.

During DNA replication, the assembly and disassembly of replication complexes and their regulators are critically regulated by Cullin-RING (Really Interesting New Gene) E3 ubiquitin ligases (CRL) [84]. The licensing factor CDT1 (Figure 3) must be degraded by CRL1^SKP2^ (at the G1/S transition) and CRL4^CDT2^ (during S-phase) through the ubiquitin proteasome pathways (UPP) driven by PCNA interactions. Dysregulation of CDT1 stability causes replication reactivation and polyploidy, resulting in cancer cell death and potential chemoresistance [85,86,87]. The degradation of CDK inhibitors such as p27, p21, and p57 during cell cycle transition is also controlled by CRL E3 ubiquitin ligase complexes, suggesting that the ubiquitin pathways can be a promising target for cancer treatment [88]. Two small molecules, pevonedistat (TAK-924/MLN4924) and TAS4464, are in clinical trials [89,90]. They suppress the activity of the NEDD8-activating enzyme, interfering with NEDD8 conjugation (neddylation) that activates CRLs as a key post-translational modification [91]. Thus, deliberately forcing RepStress in cancer cells can be a promising strategy for chemotherapy because it can be combined with the pre-existing RepStress (see Section 3.1) and the defective repair pathways of the cancer cells [5].

### 3.6. Detecting RepStress in Cancers

The RepStress leads to excessive RPA accumulation, which is recognized by ATR-interacting protein (ATRIP), leading to the recruitment of ATR. ATR belongs to the phosphatidylinositol 3-kinase (PI3K) like kinase (PIKK) family along with ATM (ataxia telangiectasia mutated) and DNA-PK (DNA-dependent protein kinase), which are also involved in the DDR and phosphorylate a number of overlapping proteins [92]. Localization of ATRIP and ATR to RPA filaments is insufficient for the activation of ATR. It requires the recruitment of the ATR activators TOPBP1 or ETAA1. While TOPBP1 is recruited by the 9-1-1 (RAD9-RAD1-HUS1) complex in response to RepStress, ETAA1 (Ewing tumor associated antigen 1) is recruited by RPA during unperturbed S-phase [93]. ATR subsequentially activates the downstream effector protein checkpoint kinase 1 (CHK1) to arrest cell cycle and stabilize replication forks [94,95]. CHK1 negatively regulates CDK activity through phosphorylation and inactivation of CDC25, while WEE1 directly inactivates CDKs by phosphorylation, and acts as a G2/M checkpoint to prevent mitotic entry of incompletely replicated DNA [96]. The DSBs generated in the context of RepStress are sensed by the Mre11-Rad50-Nbs1 (MRN) complex [33], which triggers the activation of ATM [92,97]. ATR and ATM phosphorylate the histone variant H2AX (at Ser139), which is referred to as γH2AX, and whose detection is a sensitive biomarker for DSBs and RepStress [98,99].

Detecting RepStress in tumor samples remains a challenge. Measures of RepStress including ssDNA, RPA levels bound to ssDNA, and γH2AX are widely used in experimental settings but are not optimized for use in large cohorts of clinical biopsy samples. Proteomic biomarkers to be considered for reliable immunohistochemistry include RPA1 (#IHC-00409, Bethyl labs, Montgomery, TX, USA), RPA2 (#IHC-00417, Bethyl labs), hyperphosphorylated RPA2 (S4/S8, #A300-245A, Bethyl labs), and γH2AX (S139, #07-627, Millipore, Burlington, MA, USA).

We recently developed a transcriptional profiling-based approach—the RepStress gene signature—that characterizes the cellular response to RepStress at a functional network level. The RepStress gene signature is a weighted expression signature encompassing 18 genes—*SRSF1*, *SUV39H1*, *GINS1*, *PRPS1*, *KPNA2*, *AURKB*, *TNPO2*, *ORC6*, *CCNA2*, *LIG3*, *MTF2*, *GADD45G*, *POLA1*, *POLD4*, *POLE4*, *RFC5*, *RMI1, RRM1*—derived by leveraging established cellular characteristics that portend high RepStress including amplification of *MYC* and its paralogs (*MYCN* and *MYCL*), expression of phosphorylated CHK1 (p-CHK1), and sensitivity to the CHK1 and WEE1 cell cycle checkpoint inhibitors [100]. The RepStress gene signature was recently validated as a response predictor for ATR inhibitors across a set of 14 cell lines from different tissues of origin [64]. Figure 4 also shows its predictive value also for topotecan and gemcitabine in the Broad Institute database.

## 4. Targeting the RepStress Response with Replication Checkpoint Inhibitors

Although biallelic loss of *ATR* is early embryonic lethal [101] and hypomorphic *ATR* mutations produce Seckel syndrome with severe microcephaly and growth retardation [102]; ATR inhibitors are surprisingly well-tolerated in patients [8]. Moreover, mice with Seckel syndrome crossed with p53-deficient mice are not cancer-prone [103] and ATR has been shown to support homologous recombination repair (HRR) [63,104], which provides the rationale for developing ATR inhibitor as anticancer treatments. Most relevant to the development of clinical ATR inhibitors are the findings that genetic inactivation of ATR and ATM produce a strong synergy with DNA-targeted agents such as TOP1 inhibitors [57,85,105,106] and PARP inhibitors [60,107]. The rationale for targeting ATR is that cancer cells are under RepStress, and that their survival require active replication checkpoints, namely ATR, CHK1, and WEE1, especially when the RepStress is exacerbated by DNA-targeted agents [7] (see Section 3 and Table 1). Moreover, ATR inactivation has been proposed to be synthetic lethal with ATM inactivation because of the overlapping substrates and functions of ATR and ATM, and because ATM appears inactivated in a significant proportion of cancers (10–16% of colorectal, gastric, lung and prostate cancers) [92].

Three ATR inhibitors are in advanced clinical development: Berzosertib (M6620, alias VX-970), Ceralasertib (AZD6738), and Elimusertib (BAY1895344) [8,65,66,67,108]. Additional drugs are in early development comprising M4344 and M1774 (EMD-Serono Merck, Darmstadt, Germany), ART0380 (Artios Pharma, New York, NY, USA) and RP-3500 (Repare Therapeutics, Quebec, QC, Canada) [64]. Currently, over 30 clinical trials are ongoing with ATR inhibitors alone or in combination with DNA damaging agents including nucleoside analogues, platinum-based agents, TOP and PARP inhibitors [5,6,63,64,66,109,110,111]. Broadly, these clinical trials aim to exploit: (i) the synthetic lethal interactions of the ATR-CHK1 pathway with the overexpression of oncogenes (RAS, APOBEC3A, and c-MYC), deficiency of ATM and ARID1A (AT-rich interaction domain 1A) and SLFN11 [6,63,65,85,92] using ATR inhibitor monotherapy or (ii) the dependence of high RepStress tumors on ATR, generally in combination with RepStress-inducing agents (described in Section 3.4 and Section 3.5).

The phase I trials of Elimusertib and Berzosertib provided the clinical evidence of durable single-agent antitumor activity of an ATR inhibitor in patients with advanced cancers with ATM aberrations (ATM protein expression loss and/or ATM deleterious mutation), supporting a synthetically lethal interaction between ATM deficiency and ATR inhibition [66,67]. We recently reported that targeting RepStress provides clinical benefit in cancer patients [8]. Durable regressions were observed in response to a combination of Berzosertib and Topotecan in patients with chemotherapy-resistant small cell neuroendocrine cancers (SCNCs), tumors with high endogenous RepStress [8]. Notably, tumors responding to the combination displayed marked enrichment for pathways associated with cell-cycle progression and DNA repair, including E2F target genes, the G2-M checkpoint, ATM, ATR, and Fanconi DNA repair pathways, consistent with responding tumors harboring a RepStress phenotype. Adding Berzosertib to gemcitabine improved progression-free survival compared with gemcitabine alone in platinum-resistant high-grade serous ovarian cancer, which like SCNCs exhibit high RepStress [111]. Additional promising combination trials are ongoing with platinum derivatives [66,67].

Several questions need to be considered for the further clinical development and rational use of ATR inhibitors. First, how do the ATR inhibitors differ from each other. Our recent preclinical studies showed a broad range of potency among the ATR inhibitors with M4344 being the most potent followed by Elimusertib (BAY1895344), Berzosertib (M6620), and Ceralasertib (AZD6738) [64]. *Elimusertib* and Ceralasertib are given orally while Berzosertib is given intravenously. The oral route is advantageous for avoiding infusions and repeated administrations. Yet, the intestinal absorption may vary from patient to patient and compliance with treatment can be an issue. The differential pharmacokinetics of ATR inhibitors is another important consideration. Although their plasma elimination half-life are comparable (approximately 12 h) [65,66,67,108], serum protein binding and thus free drug availability, tumor penetration/retention, and blood-brain barrier penetration may be different.

A second question is the selection of patients who are likely to respond to ATR inhibitors. In our recent preclinical study, we found that the transcriptomic RepStress signature could predict drug response. Whether the RepStress signature will closely predict clinical response to ATR inhibitors needs to be further tested. An additional established predictor is ATM deficiency, which can be viewed as synthetic lethality as ATM and ATR exert overlapping functions and share many cellular targets, thereby creating an overreliance on the ATR pathway for DNA repair and replication checkpoints [63,92,104]. Here the challenge is to confirm loss or reduction of ATM in tumor samples. Immunohistochemistry remains to be routinely applied in the clinic and scoring ATM mutations is made difficult by the large size of the ATM gene and difficulties of predicting whether a certain mutation is deleterious or a variant without functional significance [67,92,112].

A third consideration regarding the clinical use of ATR inhibitors is how to limit the toxicity to normal tissues. Although quiescent normal cells are likely to be spared by ATR inhibitors, effects on replicating cells will be dose-limiting. This practical challenge is being addressed by adjusting doses and schedules in the combinations of ATR inhibitors and DNA-targeted chemotherapies [8,65,66]. Another approach being explored in our NCI clinic is to combine ATR inhibitors with tumor-targeted DNA damaging agents, such as tumor-targeted TOP1 inhibitors (TTT is). This approach is based on our “Gap-schedule” protocol with tumor-targeted delivery of DNA damaging agents [30] (see part 4 in Figure 2). In short, the ATR inhibitor is given during the interval between the administrations of the TTT is when the normal tissues have cleared the TOP1 inhibitors, while the tumors remain loaded with the TOP1 inhibitor [30].

The clinically developed CHK1 inhibitor (SRA737) also confirmed synthetic lethal interaction with the B-family of DNA polymerases (POLA1, POLE, and POLE2) [113]. Another CHK1 inhibitor (LY2606368/prexasertib) synergistically suppressed tumor burden with DNA-damaging agents in subtypes of medulloblastoma [114]. Furthermore, prexasertib led to HRD deficiency in triple-negative breast cancer cells, thereby promoting sensitivity to the PARP inhibitor olaparib [115]. As for the ATR and CHK1 inhibitors, the clinical WEE1 inhibitor Adavosertib (AZD1775) increases unscheduled origin firings that lead to replication fork stalling, implying it could be synergistic with replicative DNA damaging agents [116]. Adavosertib also enhances the anticancer efficacy of carboplatin in patients with TP53-mutated ovarian cancer [117].

## 5. Exploiting SLFN11 as a Therapeutic Biomarker

In 2012, we discovered that SLFN11 is a dominant predictor of response to replication damaging agents widely used in cancer chemotherapy including TOP inhibitors, nucleoside analogues, platinum-based, and alkylating agents [118] (Table 1). This observation was simultaneously validated by the Broad Institute team for the TOP1 inhibitors [119], and further studies have extended the causality of SLFN11 expression to PARP inhibitors regardless of BRCA1/2 mutations [62,120].

Remarkably, SLFN11 expression is bimodal in cancer cell lines with ~50% of the NCI, CCLE and GDSC cancer cell lines expressing high *SLFN11* and the other ~50% cell lines not expressing *SLFN11* (Figure 5) [11,15,119]. A similar broad range of expression is observed in human cancers [9,11,121]. The cancers with consistently high SLFN11 expression are Ewing’s sarcoma and hematological malignancies [11,122,123]. This observation is related to the transcriptional regulation of SLFN11 by the ETS (erythroblast transformation specificity) transcription factors, and most notably FLI1 [11,122].

Epigenetic modifications, which are frequent in tumorigenesis and chemoresistance, are responsible for the lack of *SLFN11* expression in many cancer cells, resulting in chemoresistance to widely used clinical agents that target DNA replication [11,15,121,124,125,126]. Accordingly, reactivation of SLFN11 expression by epigenetic drugs targeting DNA methylation, histone deacetylase (HDAC), and histone methyltransferase EZH2 (Enhancer of zeste homolog 2) can reverse the chemoresistance of cancer cell lines that do not express SLFN11 (Figure 6A) [121,125,127]. Thus, based on the high dynamic range of SLFN11 expression and its epigenetic regulation, profiling SLFN11 status can be utilized to inform therapeutic options for precision medicine with epigenetic modulators (Figure 6A) [9,11,128].

### 5.1. Molecular Activities of SLFN11

Although SLFN11 is not activated by ATR, it negatively regulates replication fork progression in response to RepStress induced by replicative DNA-damaging agents [129]. SLFN11 is recruited by RPA on stalled forks and damaged replication sites [129,130] to hinder the CMG helicase, chromatin remodeling, and induce the degradation of CDT1 (Figure 1) [85,129,130,131]. These findings indicate that SLFN11 irreversibly blocks stressed replication forks and the firing of distant origins, which is different from the classical ATR-CHK1-mediated checkpoint pathway that transiently arrests replication [11] (Figure 1). The irreversible pathway of SLFN11 in RepStress is supported by recent findings that SLFN11 degrades stalled replication forks induced by interstrand crosslink inducers in Fanconi anemia cells, disrupts codon-specific translation process in response to DNA damaging agents, and stimulates CUL4-DDB1^CDT2^ E3 ubiquitin ligase to remove replication-related factors [85,132,133]. These molecular effects of SLFN11 have been linked to its putative helicase activity encoded in the C-terminal domain of SLFN11 (Figure 6B) [85,118,129,131]. In addition, SLFN11 has been found to act outside of the nucleus by degrading type II tRNAs and blocking the translation of ATR and ATM [133,134,135]. This ribonuclease function of SLFN11 is likely related to its N-terminal domain, which is conserved in all of human *SLFN* genes (Figure 6B) [133,136]. Recently we also reported that SLFN11 regulates proteotoxic stress and the UPR (Unfolded Protein Response) pathway outside of the nucleus [137].

### 5.2. SLFN11 as a Predictive Biomarker

Based on the findings in cancer cells-based models, the clinical exploitation of SLFN11 expression as a treatment biomarker is being actively explored. Cancer patients whose tumors show high SLFN11 expression show better chemotherapy response to PARP inhibitors, irinotecan, and temozolomide in recurrent or refractory solid tumors such as Ewing sarcoma and small cell lung cancers (SCLCs) [138,139,140]. SLFN11 is positively linked to stromal signatures of basal-like phenotype and estrogen receptor negative (ER-) type in breast cancer [141]. SLFN11 expression as a prognostic biomarker has also been confirmed in esophageal and gastric cancers after chemoradiotherapy, showing a positive correlation. Esophageal cancer patients who receive chemotherapy with nedaplatin and 5-fluorouracil and have high SLFN11 expression show a significantly longer overall survival than patients whose tumors express low SLFN11 [142]. Gastric cancer patients with lower tumor classification and stage show high SLFN11 expression, exhibiting better survival rate after platinum-based chemotherapy [143]. Moreover, chemoresistance to TOP1 or PARP inhibitors due to loss of SLFN11 can be overcome with ATR inhibitors [62,64,65] (Figure 1 and Figure 6A). By RNA sequencing of metastatic biopsy tissue, SLFN11 expression has been reported as a better prognostic marker in platinum chemotherapy than histology and genomic alterations in patients with metastatic castration-resistant prostate cancers [144]. Notably, promoter hypermethylation of SLFN11, which is a key source for SLFN11 inactivation, also predicts worse drug-response in platinum-based chemotherapy in ovarian or lung cancer patients, as well as a poor clinical outcome biomarker [125]. Yet, prospective clinical trials are warranted to establish SLFN11 status as a practical biomarker for patient selection.

### 5.3. Clinical Evaluation of SLFN11

With regards to translating SLFN11 to the clinic, reliable approaches are required to evaluate the SLFN11 status of tumor tissues and circulating tumor cells (CTCs) (Figure 6A). Studies performed at different cancer research centers have established the feasibility of immunohistochemistry (IHC)-based detection of SLFN11 in malignant and adjacent normal tissues [9,11,100,120,138,140,141,142,143,145]. SLFN11 expression by IHC is also under-investigation in CTCs from advance prostate cancer and SCLC patients [144,146].

It is important to note that SLFN11 expression in tumors is not limited to the cancer cells but can also be detected in stromal cells (see Figure 6 in [11]). IHC assessment may therefore need to specify whether the SLFN11-positive cells are cancer cells and/or stromal cells. Indeed, SLFN11 is normally expressed in lymphocytes [11,126]. Moreover, SLFN11 activity is not limited to the nucleus [133,134,135,137,147] and it remains to be determined whether cytosolic IHC staining of SLFN11 should be taken into consideration [141,148]. This is particularly relevant to the fact that human SLFN proteins including SLFN5, 11, 12, 12L, 13, and 14 have strong similarity in their amino acid sequence (Figure 6B) [11,149] and that some of these SLFN proteins only act in the cytosol [149,150]. Hence, careful examination of the epitopes is important for assessing the specificity of SLFN11 antibodies. The above consideration implies that IHC reports need to be carefully assessed to properly develop SLFN11 as a tumor biomarker (Figure 6B).

Given the highly significant correlation between SLFN11 mRNA expression and SLFN11 protein expression [118,119,121,128], RNA sequencing of patient tumor cells can be readily applied to determine the levels of SLFN11 expression. In contrast to IHC methods, profiling SLFN11 mRNA expression needs to consider the infiltrating immune cells in tumor tissues that may influence SLFN11 positiveness (see the above paragraph). Lastly, measurement of promoter hypermethylation of *SLFN11* by using DNA samples can be useful for predicting chemosensitivity and prognosis. SLFN11 inactivation in cancer cells is primarily driven by epigenetic modifications. Yet, exploitation of SLFN11 promoter methylation needs to be interpreted with caution as a large number of cancer cells also suppress SLFN11 expression by histone acetylation, which escapes from promoter methylation evaluations [15,121,151]. Nevertheless, SLFN11 promoter methylation can be used to assist the accuracy and reliability of SLFN11 inactivation determined by IHC and RNA sequencing.

## 6. Conclusions and Perspectives

Targeting DNA replication, checkpoints, and repair pathways are promising approaches for cancer treatment. A rich portfolio of drugs is in development, especially targeting ATR. Although the CHK1 inhibitors UCN-01 (7-hydroxystaurosprine), SRA737, and prexasertib preceded the ATR inhibitors, their development is lagging. It is plausible that the successful development of the ATR and WEE1 inhibitors will generate a renewed interest in CHK1 inhibitors.

The successful development of the ATR inhibitors and the use of SLFN11 as predictive biomarkers are likely to benefit from the rapid progress in precision medicine as better technologies are becoming available to map the genome and the pathways selective to cancer cells. The development of multivariate analyses appears promising and the RepStress signature is a step in that direction. Solid molecular data and detailed patient annotations should form the basis for artificial intelligence approaches that ultimately will generate therapeutic options based on individual patient characteristics.

Although we did not discuss the tumor microenvironment (TME) here, its contribution to the activity of drugs targeting the RepStress is an active area of investigation. Further investigations are warranted to elucidate the potential roles of SLFN11 in regulating and executing the immune responses. *SLFN* family genes have been discovered in the context of the immune system and they are commonly referred to as interferon-inducible genes. Therefore, elucidating the molecular connections between the immune system and the RepStress can be viewed as opening new avenues for cancer treatments.

## Figures and Tables

**Figure 1 cancers-13-04601-f001:**
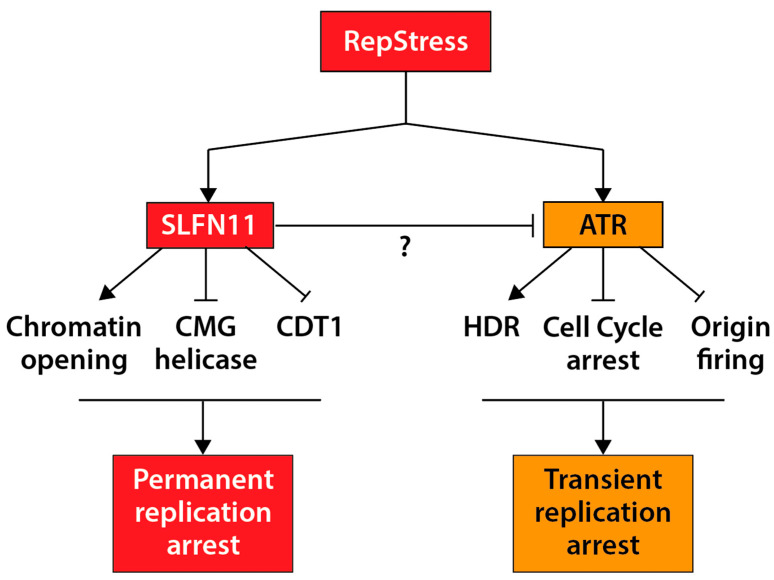
The SLFN11 and ATR pathways in response to RepStress. SLFN11 irreversibly blocks DNA replication under RepStress by promoting chromatin opening, blocking the CMG helicase complex, and promoting the degradation of CDT1 (see Section 5.1 for details). In contrast, ATR transiently halts DNA replication by arresting cell cycle and prohibiting origin firing, thereby enabling homology-directed repair (HDR).

**Figure 2 cancers-13-04601-f002:**
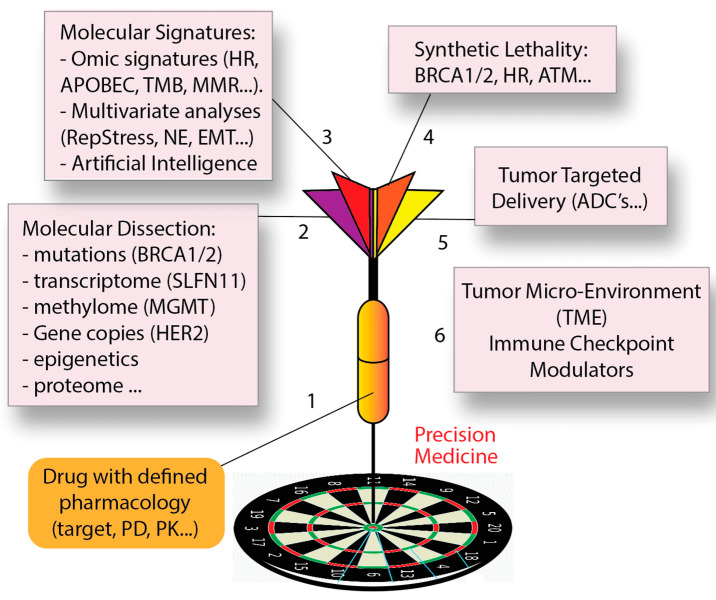
Precision medicine in the context of DNA-targeted therapies. Critical steps are: (**1**) Defining the molecular and clinical pharmacology of the drugs; (**2**) molecular dissection of tumors by multi-omics approaches; (**3**) identification of molecular signatures combining omic parameters; (**4**) determination of synthetic lethal interactions; (**5**) Targeted-delivery of drugs to cancer cells; and (**6**) adjuvant therapy by targeting the tumor microenvironment and immune checkpoints.

**Figure 3 cancers-13-04601-f003:**
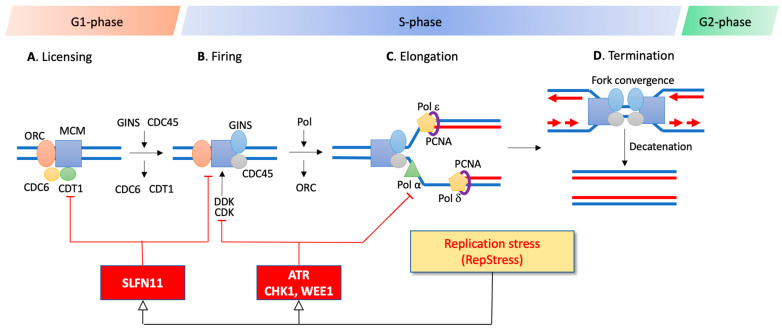
Schematic representation of the DNA replication steps in the context of the cell cycle and of the ATR and SLFN11 replication checkpoints. DNA replication is initiated by recognition of replication origins scattered along the genome. A, In G1, active origins of replication are licensed by ORC, CDC6, CDT1, and MCM2-7, forming the pre-replication complex (pre-RC). B, At the G1-S transition, the pre-RC complex is fired by activation of kinases (DDK and CDK) and loading GINS and CDC45, thereby unwinding the DNA duplex and initiating the replication by DNA polymerases. C, Active replication forks are elongated by the replication machinery including helicases, PCNA and DNA polymerases throughout S-phase. D, The replication process is terminated when the bidirectional replication forks merge. The RepStress is monitored and modulated by replication checkpoints, ATR and SLFN11, which are both recruited to RPA filaments. While ATR produces a transient cell cycle arrest allowing DNA repair, SLFN11 produces an irreversible replication arrest.

**Figure 4 cancers-13-04601-f004:**
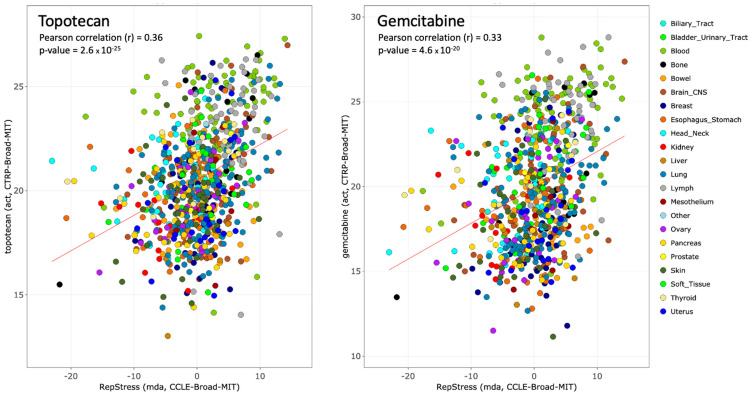
Predictive value of the RepStress genomic signature in the CCLE-CTRP Broad Institute cancer cell line databases for the TOP1 inhibitor topotecan (**left**) and the replication inhibitor gemcitabine (**right**). Data were generated from CellMinerCDB (https://discover.nci.nih.gov/cellminercdb/ (accessed on 9 February 2021)).

**Figure 5 cancers-13-04601-f005:**
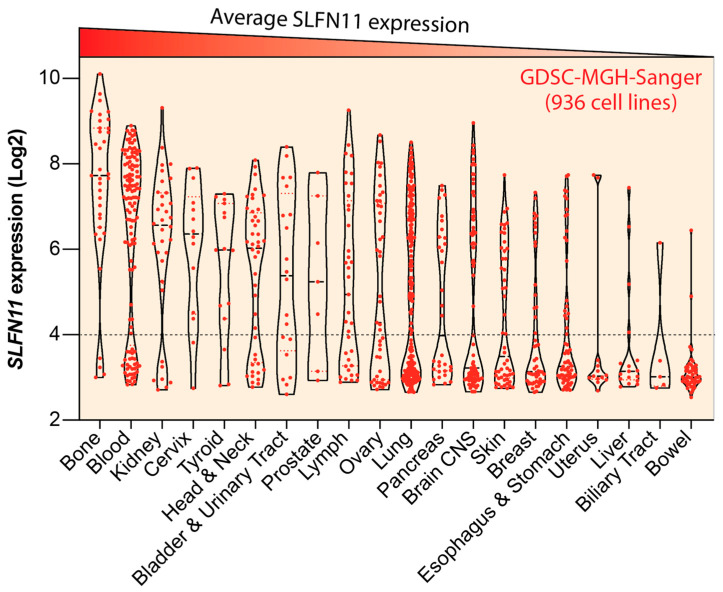
Expression status of SLFN11 across cancer cell types. Violin plot of *SLFN11* expression for the GDSC-MGH-Sanger cancer cell line dataset (https://discover.nci.nih.gov/cellminercdb/ (accessed on 9 February 2021)) (total of 936 cancer cell lines). Individual cell lines are represented as red dots. The cell lines are grouped by cancer type in decreasing order (from left to right) of SLFN11 expression. Note the bimodal expression in most tissue types. Value below 4 (dotted line) indicate background (lack of) SLFN11 expression.

**Figure 6 cancers-13-04601-f006:**
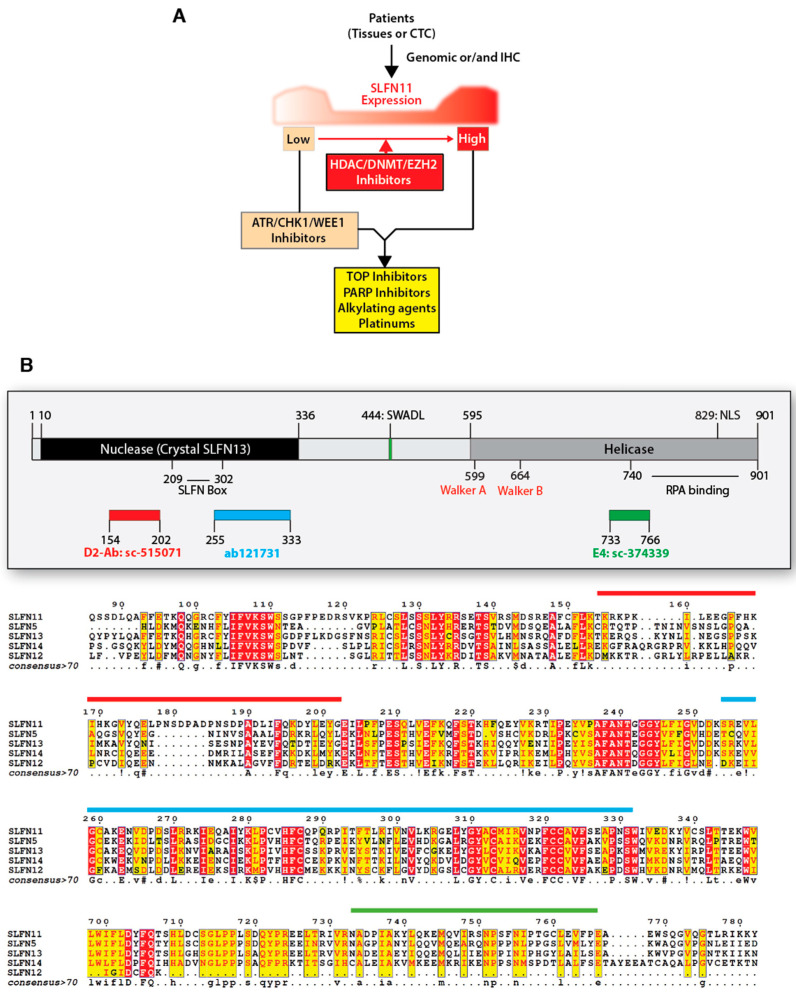
Translating SLFN11 to the clinic. (**A**) Guidance of SLFN11-coupled cancer therapy. Profiling SLFN11 expression levels in tumor tissues and circulating tumor cells (CTC) provides a predictive biomarker. SLFN11 expression can be reactivated by the inhibitors of epigenetic modulators (HDAC, DNMT, or EZH2). Low SLFN11 tumors are generally resistant to DNA damaging agents. Resistance of low SLFN11 tumors can be overcome synergistically by combination with replication checkpoint inhibitors (ATR/CHK1/WEE1 inhibitors). (**B**) Schematics of the SLFN11 polypeptide with its two main functional domains, its nuclear localization signal (NLS), its putative helicase Walker domains, and the epitopes for the commonly used antibodies. The bottom part shows the sequence alignment of human SLFNs with the targeted sequences for the available SLFN11 antibodies used for IHC detection. (D-2 and E-4: Santa Cruz Biotechnology, Dallas, TX, USA, ab121731: Abcam, Cambridge, UK).

**Table 1 cancers-13-04601-t001:** Pharmacological targets and drugs inducing replication stress.

Primary Target	Clinically Approved Drugs	Drugs in Development or Preclinical	How They Target Replication	Predictive Biomarker
Nucleoside analogs	Cytarabine Gemcitabine 5-azacytidine Decitabine		Incorporation into newly synthesized DNA blocking polymerases	SLFN11 RepStress
Ribonucleotide reductase (RNR)	Hydroxyurea Gemcitabine		Depletion of deoxyribonucleotides (dNTPs)	SLFN11 RepStress
Thymidylate synthetase (TS)	Methotrexate Pemetrexed		Thymidine depletion	SLFN11 RepStress
Dihydrofolate reductase (DHFR)	Methotrexate Pemetrexed		Thymidine depletion	SLFN11 RepStress
POLA, POLE, POLD		Aphidicolin	Polymerase arrest with chain termination	SLFN11 RepStress
POLA1		CD437		SLFN11
POLQ			Blocks MHEJ	
TLS polymerases			Blocks POL H	
DNA template MGMT	Temozolomide		06-methyl-guanine-SSB	MGMT, MMR, SLFN11, RepStress
DNA template	Nitrosoureas		DNA adducts-SSBs	
DNA template	Cyclophosphamide		DNA-DNA crosslinks	
DNA template	Cisplatin; Carboplatin Oxaliplatin		DNA-DNA crosslinks & DPC	SLFN11 HR
TOP1 trapping	Camptothecins (topotecan, irinotecan, belotecan) TTT is (Enhertu, Trodelvy	Indenoisoquinolines (LMP400, LMP776, LMP744) PLX038 CBX-12	Replication blocks (3′-DPC, SSBs) ≥ Replication run-off	SLFN11 HR RepStress
TOP2 trapping	Doxorubicin Epirubicin Etoposide mitoxantrone		Replication blocks (5′-DPC, SSBs & DSBs)	SLFN11 HR RepStress
PARP trapping (catalytic inhibition)	Talazoparib Olaparib Niraparib Rucaparib	Veliparib	Replication blocks Defective DSB repair	SLFN11 HR RepStress
ATR		Berzosertib (M6620) Ceralasertib (AZD6738)BAY1895344 M4344, M1774	Abrogate the cell cycle checkpoints as well as CHK1 and WEE1	ATM, TP53, APOBEC3B, ARID1A, MYC RepStress
CHK1		Prexasertib SRA737	Abrogates cell cycle checkpoint-activates CDKs	
WEE1		Adavosertib (AZD1775)	Abrogates cell cycle checkpoint-activates CDKs	
NEDD8 (Cullin neddylation)		Pevonedistat (TAK-924/MLN4924) TAS4464	CDT1 stabilization ≥ unscheduled origin firing	SLFN11 RepStress

## Data Availability

Not applicable.

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
