# Peer review of "Precision Oncology with Drugs Targeting the Replication Stress, ATR, and Schlafen 11"

_cancers, 2021, doi:10.3390/cancers13184601_

Round 1

Reviewer 1 Report

This review manuscript by Jo et al., and coming from the laboratory of Dr. Pommier discusses the concept of precision medicine in the context of cancer treatment by DNA-targeted drugs. After establishing the framework of precision medicine with DNA-targeted therapies in oncology, this review focuses on the signaling associated with the DNA damage response (DDR) and more specifically the replication stress response (RSR). The authors then engage in a an extensive review of approved and non-approved drugs and how they induce the RSR. Finally and most interestingly, the authors expose their vision on RSR detection in cancers and how to target it with checkpoint inhibitors and the use of SLFN11 as a biomarker for DNA-targeted therapy response prediction.

This work comes from a lab with long standing expertise in DNA topoisomerases and cancers. Although the topic is undoubtedly of utmost importance and timely, the manuscript in its current version is relatively long and could benefit from some clarifications by re-organizing and condensing some sections. Please see my detailed comments below:

  • The title and manuscript in general are relatively wordy and could benefit from some rephrasing, if possible. There are also grammatical mistakes and typos/missing words along the manuscript that should be fixed. Specifically, please clarify the sentences lines 103-104, line 137, 208, and 418-419.
  • There is not much distinction between the abstract and the introduction. In fact, the first two paragraphs of “section 2. Precision Medicine” could rather complement the intro section as the framework of this paper. Indeed, the intro does not clearly define the concepts nor addresses what the standing status of precision medicine with DNA-targeted therapies currently is, and what its rationale is. On the other hand, the abstract sounds a little vague because the critical information is too diluted. Being more specific would help getting a more forceful impact.
  • In the context of the whole review, it seems that section 3 (except for part 3.6) on DNA replication, RSR and genomic stability diverts from the story line a little too much and lengthen the review. I would focus on explaining the main ideas of Table 1 and the novelty of part 3.6. Figure 2 is not necessary in my opinion, or else could be somehow simplified and incorporated into the current Figure 4. Hopefully these modifications will also reduce the large number of references.
  • Reference 90 should rather be Bass et al., Nat Cell Biol 2016.
  • Figures and tables:
    • Fig 1 legend is too vague. Please be more specific. Only #1 is described.
    • Figure 4A and its legend are really confusing to me. Could it be possible to clarify it?
    • Maybe placing actual Figure 4B as Figure 1B would help with setting up the stage for the review and guiding the reader on the reasons why the focus is on ATRi and SLFN11?

Others/Typos:

  • Line 67: add (RepStress) after Replication Stress. What is the asterisk for?
  • Line 89: remove almost before cancer
  • Line 147: Proliferating Cell Nuclear Antigen
  • Line 510: properly

Reviewer 2 Report

This is a review entitled “ Precision Oncology with Drugs Targeting the Replication Stress and the ATR and Schlafen 11 Replication Checkpoints”, Jo and colleagues summarize advances in the uses of replication machinery as a target for therapeutic strategies. I found this to be a very thoughtful review of replication stress, SLFN11and the potential therapeutic targets for precision medicine. The review is very well-organized and well-written. I think it will stimulate research in the field and be helpful to trainees. The figures and table are very informative. I enjoy reading it and I have no additional comments or suggestions.

Reviewer 3 Report

In this review article, Jo et al emphasize the importance of targeting DNA replication, cell cycle checkpoints and DNA repair pathways with ATR inhibitors for cancer treatment. They shed the light on exploiting SLFN11 as a promising biomarker to predict the efficacy of patient-specific therapeutic strategies.

The review is very interesting and highlights recent advances in the field of cancer diagnosis and treatment where the concept of personalized and precision medicine should be of the utmost interest. The review is very well written and the figures are clear. I really enjoyed reading the paper and learned a lot from Table 1, which is nicely concise and precise.

 I have just two minor comments.

1- A small introduction to the rationale for using SLFN11 as a therapeutic biomarker is missing in the introductory section. It was detailed in paragraph 5 but not before.

2- To get a panoramic view of SLFN11 expression, authors should insert an additional figure showing the expression pattern of SLFN11 in several human cancer cell lines. (Although this has already been shown in previous references).
